# Dental emergency: Scoping review

**Karla Frichembruder**[1]*, **Camila Mello dos Santos**[1,2], **Fernando Neves Hugo**[3]

1 Center of Social Dentistry Research, Federal University of Rio Grande do Sul, Porto Alegre, Rio Grande do Sul, Brazil, 2 Graduate Program in Collective Health, Federal University of Rio Grande do Sul, Porto Alegre, Rio Grande do Sul, Brazil, 3 Graduate Program in Dentistry, Federal University of Rio Grande do Sul, Porto Alegre, RS, Brazil

☯ These authors contributed equally to this work.
* karla.frichembruder@ufrgs.br

**Data Availability Statement:** All relevant data are within the manuscript.

**Funding:** The author(s) received no specific funding for this work.

## Abstract

Part of the oral health care in the care network encompasses users in emergency cases. This study proposed mapping the determinants of the use of dental care services within the health care network to address dental emergencies within the Brazilian Unified Health System (UHS) and to verify the main gaps in the research in this area. This is a scoping review that took place in 2018 using Andersen's behavioral model as a reference. A total of 16 studies, out of 3786 original articles identified, were included and reviewed. Two reviewers independently conducted the selection process and the decision was consensually made. The mapping of the determinants revealed a greater number of enabling factors and a larger gap in the results. Greater use of the emergency service was registered by people in pain, women, adults, those from an urban area, people with a lower income, and those with less education. In future studies, primary surveys are recommended, which include all ages, and analyze different groups of needs and users that take into account the country's northern region and the different subjects pointed out by this review.

## Introduction

The majority of oral diseases are chronic and share several determinants with other chronic nontransmissible diseases. Among the different oral diseases, untreated dental caries is the most prevalent, affecting almost half of the world's population, with a negative impact on the quality of life. The pain caused by untreated dental caries affects the quality of sleep and the ability to eat, it slows growth and negatively affects social life[1,2].

Since 1988, Brazil's Unified Health System (UHS) has attempted to build a humanized care model, centered on the patient and has been coordinating services and shaping actions for the promotion, prevention and care in Primary Health Care (PHC) through the Family Health Strategy, but it has also been reorganizing other points of the care networks. The network conformation has the intent to address the multiple health care challenges in a fragmented system primarily oriented by acute conditions. Availability, access, and the ability to quickly transition between health care providers are the defining elements of a good or otherwise unsatisfactory network interface[3].

**Competing interests:** The authors have declared
that no competing interests exist.

The expansion of oral health care in Brazil since the establishment of the UHS is undeniable, with the organization of priority programmatic actions, such as the expansion of primary care, specialized dental care and support services, care provided to pregnant women and children, and emergency services[4]. The guarantee of care provided to patients in acute conditions in public services is an ethical principle found in the guidelines of the National Policies of Humanization, Primary Care, Oral Health and Emergency Care (National Policy of Emergency Attention–NPEA)[5–8]. The network structure of emergency care with settled flows intends to address acute cases according to their risk rating to provide care locally and in the appropriate time in each case[9]. It is expected that most cases of dental emergencies will be treated in the PHC or Secondary Emergency Care, with the focus of hospital care being cases with greater risk of life. After emergency care, the patient is expected to be referred for continuity of care at one of the scheduled care points. It is recommended that scheduled specialized dental care be performed exclusively by reference. During the previous decade, there was an increase in this point of care due to the initiation of implantation at dental specialty centers (DSC).

The monitoring of the development of the care network encompasses several elements, such as the operationalization of the government system, population characteristics, operational structure and model of care. In turn, the monitoring of the use of network services requires the extension of the scope of analysis, since the use is shaped by the *"interaction of behaviors of individuals and professionals that lead them through the health system"*[10]. The mapping of determinants of the use of dental emergency services as part of the UHS is decisive for the analysis of the situation, for the planning of actions of intervention aimed at improving access and quality of care and represents a research gap.

In order to understand the different factors that may influence the use of network health services by users in dental emergency, this study aims to map the determinants of use of the emergency care network (ECN) of the UHS and verify the main research gaps. The scoping review in the Brazilian context is justified by the influence of the organizational model on the use of services.

## Material and methods

Scoping reviews are a type of knowledge synthesis that systematically maps evidence on a specific subject matter, identifying key concepts, theories, sources of evidence, and research[11]. This scoping review follows the five steps proposed by Ashley and O'Malley[12]: (1) identifying the research question, (2) identifying relevant studies, (3) selecting studies, (4) collecting data, and (5) mapping, summarizing and describing the results[12].

### Theoretical model

The theoretical model involves understanding how the use of services in the network occurs and how the factors of the behavioral model modulate access to health services. This means obtaining proper care at the right time and place to promote better health outcomes. This model is intricate and multidimensional and has been improved over the years. The model bases itself on the fact that improved access to care is more properly addressed and explained through the relationship between predisposing, enabling, needs, health behaviors and outcomes and considering contextual and individual factors[13].

### Research question

The topic of interest was dental emergencies and the research question was the following: what has been studied on the use of the dental emergency care network in Brazil's UHS public

services? The question encompasses the concept of emergency in dentistry, user-related factors, as well as the organization of the ECN, its components and organizational principles. The pre-established criterion of inclusion was to be an article on the subject of dental emergency care in the context of Brazilian public services.

## Research and study selection

In order to build the research strategies, an adapted version of the PECO strategy was adopted (P: patient, E: exposure, C: comparison, O: outcomes), turning into the PEC, in which "P" means the population (users), "E" means exposure of interest (dental emergency), and "C" means the context (health services)[14].

The health descriptors and the combinations used to build the strategies were the following: "emergencies", "emergency", "oral health", "dentistry" and "health services" with Boolean operators such as "AND" and "OR". The search was carried out in the Medline (PubMed), Embase, Web of Science and Scopus databases from their beginning until September 2018. The descriptors summarized in Medline were: (((((((((((((emergencies OR emergenc*) OR urgenc*)) OR ((out of hours) OR out-of-hours))) AND (("oral health") OR dent*))) AND (((((((((health services OR public health dentistry]) OR after-hours care OR "dental care") OR emergenc* dental service) OR emergenc* dental care) OR "oral care") OR "dental services")))))))))); in Embase: ('out-of-hours' OR 'out of hours' OR 'emergen*' OR 'urgen*') AND ('oral health' OR 'dental') AND ('emergency health service' OR 'out-of-hours care' OR 'emergency care' OR 'emergency care'); in Web of Science: (((emergencies/ OR "urgen*dental" OR "emergen* dental") OR ("out-of-hours" OR "out of hours" OR "unscheduled")) AND (dental care/ OR dental health services/ OR "dental care" OR "dental service*" OR "public health dentistry" OR "dental after-hours care")); in Scopus: (TITLE-ABS-KEY ("emergencies") OR TITLE-ABS-KEY ("emergenc*") OR TITLE-ABS-KEY ("urgenc*") OR TITLE-ABS-KEY ("out of hours") OR TITLE-ABS-KEY ("out-of-hours") AND TITLE-ABS-KEY ("oral health") OR TITLE-ABS-KEY ("dent*") AND TITLE-ABS-KEY ("health services") OR TITLE-ABS-KEY ("public health dentistry") OR TITLE-ABS-KEY ("after-hours care ") OR TITLE-ABS-KEY ("dental care") OR TITLE-ABS-KEY ("emergenc* dental care") OR TITLE-ABS-KEY ("emergenc* dental service") OR TITLE-ABS-KEY ("oral care") OR TITLE-ABS-KEY ("dental services")). Also, the search was conducted in the gray literature using the "Google Scholar" search engine.

Titles and abstracts were read and analyzed to identify those potentially eligible for the study. The selected studies were fully read by two independent reviewers to confirm the relevance when taking into consideration the review question and, when relevant, to extract the data deemed interesting.

After the completion of the search and analysis processes, the following exclusion criteria were established: published before 1990, having as a referencing point the fact that that was the year of enactment of Law 8080, which rules on the organization of health services; abstracts and articles published as part of meetings; and studies in hospitals.

## Data collection, summarization and presentation of results

The data extracted were the author, year of publication, journal, emergency concept, objectives, methodology (setting, design, population/sample, duration, outcome and exploratory variables) and results. The data were organized into Excel spreadsheets. The studies were classified according to the Emergency Care Network in PHC, DEC and ECN. The term DEC was used to take into consideration different terminologies found for specialized dental emergency services; in turn the term ECN was used to identify studies involving both points of the

emergency network. DEC are intermediary services that exclusively attend emergencies, supporting this service in the PHC and reducing the hospitalization of dental urgency in the hospital, which should refer care to the PHC, DSC, or hospitals according to the needs of the people. It includes dental care in emergency medical services that can be qualified for 24-hour care, offering beds for short-term prehospital care, in this case, receiving its own financing according to the fulfillment of pre-established goals. The studies were grouped according to the age of the participants, and studies with participants aged 20 years or more were grouped into adults and old adults. The results were categorized according to the components of the behavioral model, similar to the methodology used by Worsley et al., which evaluated access to dental emergency services[15]. Among enabling factors were the specific aspects of the Brazilian model, which are related to organization and financing that have an influence on the universal and comprehensive access to care in the network. From this standpoint, the variables of the studies that attempted to assess the perception and agreement of the professionals and managers/coordinators about the service were distributed considering the professionals and managers, and kept as capacitors, since they were understood as the evidence of the service organization; meanwhile, as observed by Worsley, there was the possibility of including them in other fields of the model. Therefore, 5, 8 and 20 variables were grouped as interfaces between PHC and DEC (health care network design, levels of health care, comprehensiveness, integration-interdependence-communication, streams of care), perspective of professionals (type of oral conditions attended, treatments, reference to hospitals, knowledge required for action, service orders, completion of treatment, more frequent type of urgency, reception and risk classification, work overload, clinical and pharmacological guidelines, referral system, continuing education, resources, patient profile, patient admission form, continuity of care, medical records, gratuities, time and attendance monitoring and managerial meetings), and perspective of managers (waiting time, structural conditions, patient admission form, professional satisfaction, social control, production goals, patient satisfaction, reference system). In health behavior, the uses of dental floss and tooth brushing were grouped into oral hygiene. Regarding the use of personal health services, 14 variables were arranged together as use due to dental emergency (difficulty in accessing dental care, emergency care as first choice to access PHC, annual trend of care at DEC, emergency service as first access to dental care, PHC or DEC as first choice for dental emergency care, return to the dental emergency service for the same problem, comparison of type of dental emergency care in different services, time since last dental appointment, unresolved complaints and abandonment).

### Protocol and registration

The scoping review adheres to the Joana Briggs Institute Coping review protocol guidelines. The protocol was registered by the protocols.io (dx.doi.org/10.17504/protocols.io.8nshvee) [16].

### Ethical considerations

This study relied on secondary data analysis, which is available in database of scientific literature and, therefore, it did not require submission to the Research Ethics Committee.

## Results

The study encompassed a total of 4297 articles after the removal of duplicates and, of these, 17 studies were included. The flowchart presents the selection of publications (Fig 1).

The time needed to carry out the study, from the research project to the completion of the article, was of approximately five months.

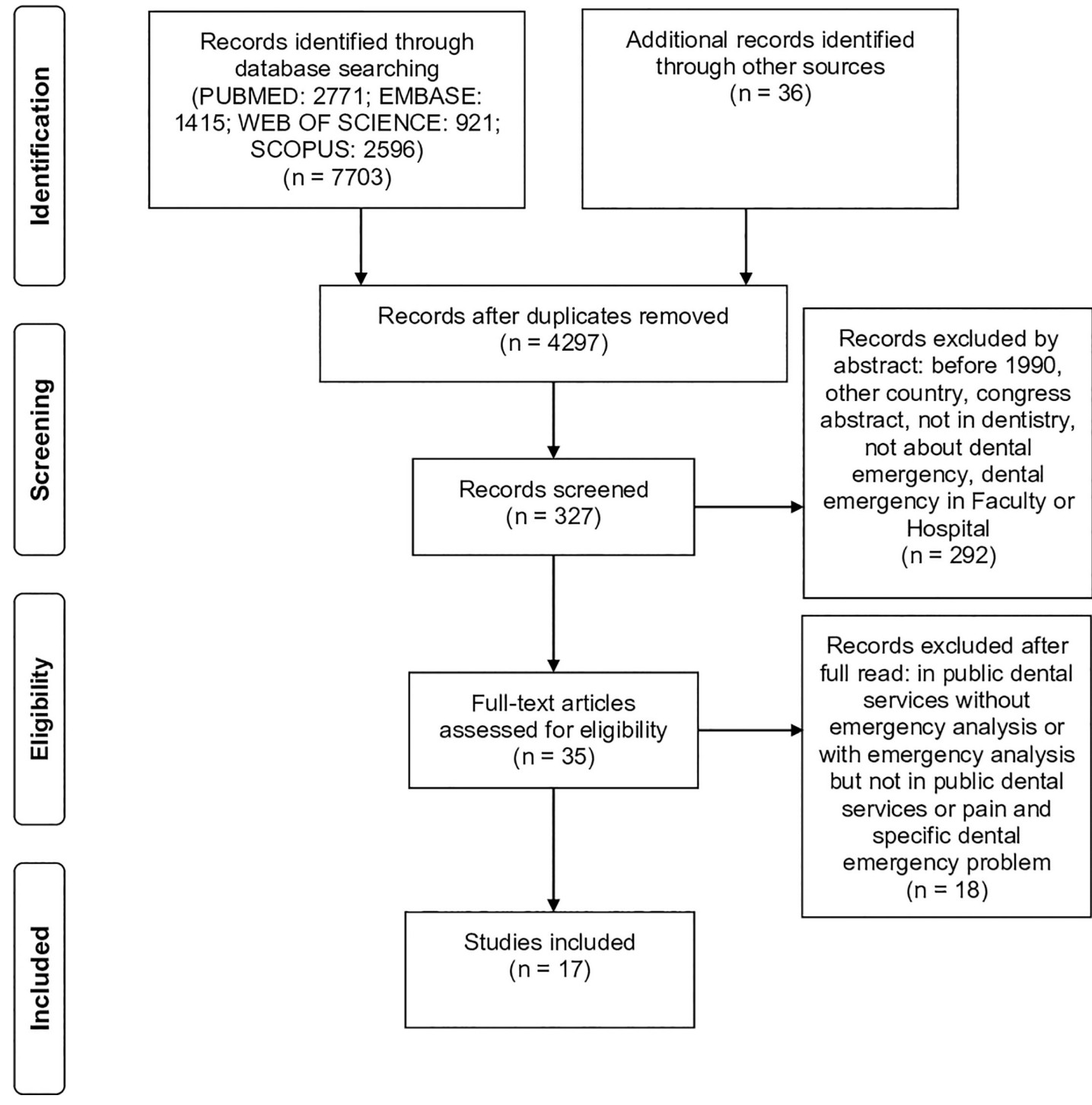

**Fig 1. Scoping review flowchart.**

## Study descriptions

Of the 17 studies included, four (23.5%) were part of the PHC, ten eleven (64.7%) of the DEC, and two (11.8%) involved the ECN. No publications were found from the country's northern

region, publications on DEC are distributed among services located in the other 4 regions of the country, articles related to PHC come solely from services located in the south and southeast regions, while publications involving the ECN come from south and northeast regions.

Among the primary studies, sixteen were quantitative (94.11%), and among them 8 (50%) were descriptive and 8 (50%) were cross-sectional. The data were secondary in 9 (56.3%), and the remaining used questionnaires. Of the 16 quantitative studies, 11(68.8%) had the individual as the unit of analysis.

## Emergency concept

Four articles (23.5%) presented three concepts of emergency: "Urgency is any immediate treatment that alleviates the patient's discomfort who is not at risk, while emergencies are serious occurrences, in which the patient requires quick care, since there is a risk of life involved"[17]; "a dental emergency is associated with immediate measures whose target is to alleviate the painful, infectious and/or aesthetic symptoms of the oral cavity"[18,19];.and "the dental emergency service can be defined as the care provided to patients with oral issues that interfere with their lives or organ functioning"[20].

## Behavioral model for the use of health services

The mapping of determining factors on the use of services, according to the included studies, is showed below (Fig 2).

Among the 154 occurrences of determining factors, the enabling factors were the most frequent (32.4%), and the outcome category was the least studied (1.9%). The research gaps are showed below (Fig 3).

## Predisposing factors and gaps

Among the predisposing factors, gender and age were the variables that appeared most frequently in the studies. The female gender was the most frequent (7/11, 63.6%) [17–27]. The studies that included all ages or those above 13 years old (n = 8/10, 66.6%) highlighted adults as the most prevalent; in studies up to 19 years old (n = 3), the highest frequency was found between 10 and 14 years[17–27]. The comparative study indicated a higher prevalence of DEC use by teenagers in comparison to children[23]. The proportionality between gender and age and the population without emergent needs, or the ascribed or reference population, was not addressed.

A low number of studies used the variables of origin, ethnicity and health predispositions. An inverse relationship between education and emergency use was observed in both points of the network (3/4, 75%)[18,22,26], one study refers to this inversion through maternal education[22]. In PHC and DEC, there was greater use by married individuals, the majority of whom were employed or autonomous (n = 2)[20,23]. In PHC, an average of two children was found, as well as double or triple shift jobs (n = 1)[22]. In DEC, the majority of those attended originated from the service's coverage area (n = 2)[21,24]; there is a higher chance of use by people from neighborhoods deemed as having a greater social exclusion rate (n = 1)[26], and rural residents accessed it less often (n = 2)[17,27]. In the contextual analysis of the PHC, the group of services with the greatest ratio of dental emergency was not the one whose area was the most vulnerable to health (n = 1)[28]. The health predispositions described the fear of going to the dentist (n = 1)[22], depression (n = 1)[22], allergies (n = 2)[19,28] and systemic health conditions, with hypertension being the most frequent (n = 1)[18]. There were several apparent research gaps addressing beliefs, knowledgeability about the service, trauma-related issues and studies conducted in specific groups such as disabled and older adults.

| Predisposing (43) | Enabling (50) | Needs (17) | Health behaviors (41) | Outcomes (3) |
|---|---|---|---|---|
| **Demographic** | **Health policy** | **Perceived** | **Personal health practices** | **Consumer satisfaction** |
| Age (11) [17-27] | Knowledge about municipal policy and DEC re-organization strategy (2) [30,31] | Complaint (9) [17-19,21,22,24,26,27] | Oral hygiene (2) [18] | Infrastructure (1) [26] |
| Gender (11) [17-27] | Interface between PHC and DEC (5) [31] | Perception of absence of need (1) [26] | Self-medication (3) [17,21,27] | Waiting time (1) [26] |
| **Social** | **Financing** | **Evaluated** | Alcohol consumption (1) [18] | OHT (1) [26] |
| Occupation (2) [22,18] | Public resource management (1) [30] | Risk rating (2) [17,27] | Smoking (2) [18,22] | |
| Education (3) [18,22,26] | Income (2) [22,26] | Diagnosis of oral disease (4) [18,21,22,28] | **Use of personal health services** | |
| Maternal education (1) [22] | Health plan (1) [26] | No dental emergency need (1) [24] | Due to dental emergency (15) [17,19,21-27] | |
| Marital status (2) [22,26] | Vehicle (1) [26] | | Day or shift (4) [17,24,25,27] | |
| Number of children (1) [22] | **Organization** | | Month (2) [24,27] | |
| Index of vulnerability (1) [28] | Expansion of the PHC (1) [29] | | In elective care (1) [19] | |
| From urban/rural area (2) [17,27] | Goal to dental emergency (1) [29] | | **Process of medical care** | |
| From area of coverage by DEC or not (2) [21,24] | Coverage by PHC (2) [28,29] | | Treatment completed (2) [18,28] | |
| From area of social exclusion or not (1) [25] | Coverage by OHT (2) [25,28] | | Type of treatment (3) [19,20,25] | |
| Ethnicity (1) [26] | DEC availability in administrative region (1) [23] | | Problem-solving (3) [17,19,26] | |
| **Predisposing conditions** | Distance from PHC service and DEC (1) [23] | | Reference in the HCN (2) [17,28] | |
| Fear (1) [22] | Perspective of professionals about DEC (22) [30,32,33] | | | |
| Health conditions (4) [17,18,22,27] | Perspective of managers about DEC (8) [30] | | | |

**Fig 2. Determinant factors of the use of dental emergency services.**

| Predisposing | Enabling | Needs | Health Behaviors | Outcomes |
|---|---|---|---|---|
| Aspects related to violence and accidents and dental emergency. | NPEA impact on the access | Assessment according to subjective and social indicators of oral health | Adherence to the treatment protocol | User satisfaction |
| On oral health and dental care | Economic studies on health and access to emergency dental services | Agreement between professionals and users | Ethics in dental emergency | Impact of dental emergency on quality of life |
| Expectations about emergency care | Financial impact of a dental emergency for the user | Impact on the emergency need in populations with and without water fluoridation | | Impact of previous traumatic experience |
| Cultural values and norms of the population that access emergency dental services | Use of the service before and after organizational changes (adhesion to protocols, risk classification, training) | Agreement between professionals, managers and coordinators from different points of the RUE | | Studies that relate the point of care and access to emergency through measures of quality of life |
| Changes in the stance of professionals in the organization of emergency care | Use of services after changes in the ECN | | | |
| Prevention and management of dental trauma | | | | |
| User knowledge on the service Disabled people | | | | |
| Anxiety | | | | |

**Fig 3. Gaps in the dental emergency research.**

## Enabling factors and gaps

There was a declining ratio between the income of users and the use of emergency services at both points of the network (n = 2)[22,26]. A study conducted in the DEC reported that most users do not have health insurance and travel by bus to the emergency[26]. When analyzing the expansion of the oral health team (OHT) in the PHC network over a three-year period, one study noticed that, in relation to the total number of appointments, there was a statistically significant reduction during the studied period that was justified by the year of greatest expansion. Nevertheless, the monthly variation was high and, although there was a reduction in the total number of emergency visits, in the vast majority of the months studied, the target of less than 20% of the total number of appointments was not reached[29]. In the ecological study, the differences between the group of services with the highest ratio of emergency and preventive procedures could not be explained by the population coverage provided by the family health team and the ESB[28].

Information on health policy, funding and organization of the ECN and the DEC is limited to a scarce number of studies with high variability in the variables gathered, but their results converge to deficiencies in the organization of the ECN. One study notes that most managers are unaware of policy updates, while others barely engage in financial planning and execution [30,31]. It was reported that most managers are unaware of objectives, they acknowledge access to the DEC by free demand, there is no waiting time science, nor one regarding the level of user satisfaction, but there is a record of criticism and suggestions provided by the user and they say that they take into account the professional satisfaction and user suggestions, while most do not engage in the Municipal Health Council (n = 1)[30]. The managers guaranteed the presence of at least one piece of equipment ready to be used, having been subjected to preventive maintenance (n = 1)[30]. There is evidence of acknowledgment of the role of each of the points among network professionals (n = 1)[31], but without any communication, protocols and reference flows (n = 2)[30,32], and with poor recognition of the lines of streams of care in the DEC (n = 1). Studies conducted in the DEC reported diverging opinions among professionals about some activities to be performed, procedures and which to refer[32,33], while there was consensus on spontaneous demand access and on the use of the medical record, science dedicated to accommodation failures and the system of risk rating, referencing, continuing education and protocol with clinical and pharmacological guidelines, as well as failures in infrastructure resources. Additionally, professionals confirm the control of workload and additional workload for the night shift[32,33]. The profile of the emergency user outlined by the professional confirms the predisposing characteristics and needs obtained in this review. The majority affirms that the user does not have a referral document, but affirms that they guide their own search under the continuity of care[30]. Most professionals acknowledge the highest contributions to their practice in graduation and in-service experience (n = 1)[32]. There are indications for research on the effect of NPEA on the ECN development and changes in access after organizational changes in the care network, related to health economics and related to the user.

## Health needs factors and gaps

The need perceived by the majority of users that led them to use the services was pain (n = 7) [17–19,22,24,26,27]. One study showed that posttraumatic injury complaints were more frequent in men and there was a noteworthy difference by age group, in which the highest prevalence of trauma and posttraumatic injuries was 0 to 5 years[24]. A share of DEC users acknowledges that they do not have an emergency need (n = 1)[26]. One study reported pain history for 9 days or less before DEC care[21]. The comparative study mentioned emergency as the reason for the first access to oral health for a portion of the population up to 17 years

old, in which adolescents are more prone to entry via DEC[23]. The identified gaps are related to the use of subjective and social indicators, the agreement between professionals and managers and the analysis of environmental contexts.

## Health behavior factors and gaps

The health practices described were oral hygiene (n = 1)[18], self-medication ((n = 2)[17,27], smoking (n = 2)[18,22] and alcoholism (n = 1)[18]. The majority of medical records did not contain information on self-medication, among which they reported low use, with analgesics being the most frequent. One study reported that most medications were taken without guidance from a healthcare professional, with sodium dipyrone being the most commonly used medication[21]. A study in the PHC setting described that most users did not experience difficulties in accessing and had already used the service for emergency-related matters, and the time between the perception of the need and the use of the service was seven days[22]. Regarding DEC, one study reported that the ratio of people attended in the estimated population did not differ over a three-year period[26]. Another study affirmed that just under one third of those attended declared that they failed to access the PHC due to infrastructure issues, the lack of openings or the absence of the medical specialty required[25]. For the majority of users, the time from the onset of symptoms to the use of care services was two days[26]. A small ratio is found in dental care; most use the UBS or health insurance[19]. The results regarding greater demand for the service depending on the shift, day and month were divergent (n = 4) [17,24,25,27], but there seems to be a relationship between the shifts used and the age group, where older people tend to use them in the morning, children in the afternoon and teenagers and young adults at night. The share of people who use the DEC and were not attended was similar in the three studies, with less than 3%[17,24,27]. The comparative study found that the prevalence of first access to the system through emergency via DEC was significantly higher in adolescents than in children. The majority of participants spent more than one year without any dental appointment, and a minority used the service previously as a matter of emergency [23]. In relation to treatment, restoration and extraction are the most frequent procedures (n = 3)[19,20,25]. From the contextual standpoint, in the PHC, the probability of being part of the group of services with the highest emergency ratio was associated with having the treatment completed for more than 3 teeth with cavities or indication of tooth extraction[24]. Regarding the problem-solving abilities in the DEC, one article noted that most of the complaints were solved and another reported that the majority of the treatments were not fully operative[17,19]. A study conducted in the DEC noted that the majority of those attended do not need to be referenced for programed care in specialized services[17]. No publications were found on the use and adherence to treatment or referral protocols, or on the needs related to continuing education or discussion of care from an ethical standpoint.

## Outcome factors and gaps

As for the outcome component, a publication presented the perception of the users regarding the DEC service. Facilities, information, cleaning and signage, waiting time and care provided by the ESB were assessed as good, with room for improvement, particularly for waiting time and care provided[26]. No studies were found that assessed the perception of postcare health and quality of life.

## Discussion

The mapping of the determining factors of the use of emergency services provided an overview of the evidence, the reflection on variables to be included in future studies and a wide array of

research topics that may lead to a better understanding of the determinants of the use of emergency dental services. There are a short number of studies involving the ECN and the PHC, and there are research gaps in the northern region of Brazil. There were a considerable number of descriptive studies, variability between the categories studied and diversity of exploratory variables that make comparisons more difficult, but they nonetheless extend the perspective on the subject matter.

The concepts found are related to the perceived need and to the organization of care, since they refer to care or service; they include the notion of time and relief of symptoms, illness and issues that interfere with the life of the user. The concept of life-threatening is the differentiator and promptly indicates the need for immediate care in tertiary care. Nevertheless, there are a scarce number of publications that conduct conceptual reflections on dental emergency. The presence of some level of disagreement regarding the activities, procedures and reference found in the DEC does not seem to reflect the conceptual range of emergency care and suggests the presence of tension in the team that may constitute barriers to access and indicate the need for improvement in the work[34].

The absence of some elements emphasized in the model can be partially explained by the restriction of data information from the Brazilian information system. Notwithstanding, the results found are in line with those of a systematic review of the inequity in access to oral health services, which, in South America, has revealed that the opportunities for access are lower for men, ethnic minorities, and rural inhabitants and distinguishes that access is greater for those with a lower educational level and income, since, in the emergency service, individuals with a lower educational level, lower income and diminished access to health plans were the ones that accessed it the most[35].

The international literature provides several reports that separate studies based on traumatic and nontraumatic dental emergencies[36–39], but not a single Brazilian study used this division, probably because of the differences between the organization of health services, since a large share of these articles refer to care in outpatient clinics and some involve care provided by a medical professional. Nonetheless, there was an analysis of traumatic events that corroborated the greater frequency of trauma in men and younger boys[40]. The prevention of trauma is difficult, and the approach varies, since the cause is related to risk factors according to age, accidents, sports and violence[40]. For instance, studies involving day-care centers and public schools show a low level of knowledge on dental trauma cases, evidencing a research gap in relation to the PHC[41,42]. There were no studies on anxiety in the ECN, and two studies in dental school services reveal that an important portion of those seen in their emergency services has a high degree of anxiety, which is higher in women and is related to previous traumatic events[43,44].

The limitations of this scoping review are the exclusion of abstracts from events and theses, dissertations and monographs, which may have caused the omission of some relevant studies. The high variability among the studied age groups in the DEC can be a confounding factor.

Studies on the care network and the integration of the PHC with the Dental Specialties Centers (DSC) presented some results similar to those in the PHC and DEC, such as failures in continuing education and reduced engagement in participation forums. Although there are also weaknesses identified in services and in the interface, they seem to register better results when it comes to the identification of objectives, the presence of protocols and reference flows [45,46]. The policies that involve secondary care in the care network, the DSC and the DEC are recent, but they have occupied different positions, since, even though both are based on the National Oral Health Policy, the regulations of the DEC services are associated with the NPEA, whose priority is not dental care[7,47,48]. The difference between having specific financial incentive rules, implementation, monitoring and assessment, as well as the

involvement of different sectors in the planning, training and monitoring of the DSC, may serve as an explanation for the better results in comparison to when these are related to a broader policy.

## Conclusion

To improve access and the quality of oral health care in Brazil, it is important to identify the determinants of the use of emergency dental services. The results converge to accumulated needs related to the aggravation of chronic oral diseases with painful symptomatology in users who are subjected to worse socioeconomic conditions and they appear to differ from the determinants of use by programmed demand. There is an evident need for improvement in each point of the ECN and in its interface, such as improvements in accommodation, assimilation of risk rating, definitions of protocols and reference flows, which require the involvement of professionals and managers in every network sector. This review also contributes to the reflection on variables, subject matter and research designs that must be taken into account in the planning of new studies, as there is a need for further research efforts on the performance of services and the care network and effectiveness of this sort of care in dental emergencies.

## Supporting information

**S1 Fig. Scoping review flowchart.**
(PDF)

**S2 Fig. Determinant factors in use of dental emergency services.**
(PDF)

**S3 Fig. Gaps in dental emergency research.**
(PDF)

**S4 Fig. Prisma checklist Scoping Review.**
(PDF)

## Author Contributions

**Conceptualization:** Karla Frichembruder, Camila Mello dos Santos, Fernando Neves Hugo.

**Data curation:** Karla Frichembruder, Camila Mello dos Santos.

**Formal analysis:** Karla Frichembruder, Camila Mello dos Santos.

**Investigation:** Karla Frichembruder.

**Methodology:** Karla Frichembruder, Camila Mello dos Santos, Fernando Neves Hugo.

**Project administration:** Karla Frichembruder, Fernando Neves Hugo.

**Visualization:** Karla Frichembruder.

**Writing – original draft:** Karla Frichembruder, Fernando Neves Hugo.

**Writing – review & editing:** Karla Frichembruder, Camila Mello dos Santos, Fernando Neves Hugo.

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
