## [Decision Letter · Decision Letter 0]

23 Oct 2019

PONE-D-19-23071

Dental emergency: scoping review

PLOS ONE

Dear Mrs Frichembruder,

Thank you for submitting your manuscript to PLOS ONE. After careful consideration, we
feel that it has merit but does not fully meet PLOS ONE’s publication criteria as it
currently stands. Therefore, we invite you to submit a revised version of the
manuscript that address

We would appreciate receiving your revised manuscript by Dec 07 2019 11:59PM. When
you are ready to submit your revision, log on to https://www.editorialmanager.com/pone/ and select the 'Submissions
Needing Revision' folder to locate your manuscript file.

If you would like to make changes to your financial disclosure, please include your
updated statement in your cover letter.

To enhance the reproducibility of your results, we recommend that if applicable you
deposit your laboratory protocols in protocols.io, where a protocol can be assigned
its own identifier (DOI) such that it can be cited independently in the future. For
instructions see: http://journals.plos.org/plosone/s/submission-guidelines#loc-laboratory-protocols

We look forward to receiving your revised manuscript.

Kind regards,

Marta Pascual

Academic Editor

PLOS ONE

**Journal Requirements:**

2. Our editorial staff has assessed your submission, and we have concerns about the
grammar, usage, and overall readability of the manuscript.  We therefore request
that you revise the text to fix the grammatical errors and improve the overall
readability of the text before we send it for review. We suggest you have a fluent,
preferably native, English-language speaker thoroughly copyedit your manuscript for
language usage, spelling, and grammar.

If you do not know anyone who can do this, you may wish to consider employing a
professional scientific editing service.  

Whilst you may use any professional scientific editing service of your choice, PLOS
has partnered with both American Journal Experts (AJE) and Editage to provide
discounted services to PLOS authors. Both organizations have experience helping
authors meet PLOS guidelines and can provide language editing, translation,
manuscript formatting, and figure formatting to ensure your manuscript meets our
submission guidelines. To take advantage of our partnership with AJE, visit the AJE
website (http://learn.aje.com/plos/) and enter referral
code PLOS15 for a 15% discount off AJE services. To take advantage of our
partnership with Editage, visit the Editage website (www.editage.com) and enter referral code PLOSEDIT for a 15% discount
off Editage services. If the PLOS editorial team finds any language issues in text
that either AJE or Editage has edited, the service provider will re-edit the text
for free.

Please note that PLOS ONE does not copyedit accepted manuscripts and that one of our
criteria for publication is that articles must be presented in an intelligible
fashion and written in clear, correct, and unambiguous English (http://www.plosone.org/static/publication#language). If the language
is not sufficiently improved, we may have no choice but to reject the manuscript
without review.

**Comments to the Author**

1. Is the manuscript technically sound, and do the data support the conclusions?

Reviewer #1: Yes

Reviewer #2: Yes

Reviewer #3: Yes

2. Has the statistical analysis been performed
appropriately and rigorously? 

Reviewer #1: N/A

Reviewer #2: N/A

Reviewer #3: N/A

3. Have the authors made all data underlying the
findings in their manuscript fully available?

Reviewer #1: Yes

Reviewer #2: Yes

Reviewer #3: Yes

4. Is the manuscript presented in an intelligible
fashion and written in standard English?

Reviewer #1: Yes

Reviewer #2: Yes

Reviewer #3: Yes

5. Review Comments to the Author

Reviewer #1: - More search engines are recommended, as you stated The search was
carried out in the Medline (PubMed), Embase and Web of Science data-bases
.......

- More statistical analysis is required to present your results clearly.

Reviewer #2: This is a review article, so there is no statistical concern. Author has
describe the Emergency Care Network (ECN) of the country very clearly and advised to
improve ECN net work to understand dental emergencies.

Reviewer #3: In general, the manuscript followed a clear and well written format that
was easy to follow and understand. However, there are minor grammatical errors which
I have highlighted in the attached document along with this form.

6. PLOS authors have the option to publish the peer
review history of their article (what does this mean?). If published, this will
include your full peer review and any attached files.

If you choose “no”, your identity will remain anonymous but your review may still be
made public.

**Do you want your identity to be public for this peer review?** For
information about this choice, including consent withdrawal, please see our
Privacy Policy.

Reviewer #1: No

Reviewer #2: No

Reviewer #3: Yes: Sakina Abbasher

---

## [Author Response · Author response to Decision Letter 0]

2 Dec 2019

Academic Editor:

I have incorporated all your suggestion into my revision. They were very helpful.
Thank you. (protocols.io and registered as: dx.doi.org/10.17504/protocols.io.8nshvee; the manuscript was edited and
certificated by American Journal Expert) 

Revisor 1:

I have incorporated your suggestions into my revision. Only descriptive analyses were
provided because scoping reviews do not make other statistics analyses. Data
synthesis by metarregression is a common feature of Systematic Reviews with
meta-analysis, but this was not the case of the present study. Thank you.

Revisor 3:

I have incorporated all your suggestion into my revision. They were very helpful.
Thank you.

to reviewers.docx
---

## [Editor Report · Decision Letter 1]

24 Jan 2020

Dental emergency: scoping review

PONE-D-19-23071R1

Dear Dr. Frichembruder,

We are pleased to inform you that your manuscript has been judged scientifically
suitable for publication and will be formally accepted for publication once it
complies with all outstanding technical requirements.

With kind regards,

Gururaj Arakeri, MDS, Ph.D, FIAOO

Academic Editor

PLOS ONE

---

## [Editor Report · Acceptance letter]

7 Feb 2020

PONE-D-19-23071R1 

Dental emergency: scoping review 

Dear Dr. Frichembruder:

I am pleased to inform you that your manuscript has been deemed suitable for
publication in PLOS ONE. Congratulations! Your manuscript is now with our production
department. 

With kind regards,

on behalf of

Dr Gururaj Arakeri 

Academic Editor

PLOS ONE